# VEGF in Tears as a Biomarker for Exudative Age-Related Macular Degeneration: Molecular Dynamics in a Mouse Model and Human Samples

**DOI:** 10.3390/ijms26083855

**Published:** 2025-04-18

**Authors:** Seyed Mohamad Mehdi Moshtaghion, Filippo Locri, Alvaro Plaza Reyes, Flavia Plastino, Anders Kvanta, Maria Jose Morillo-Sanchez, Enrique Rodríguez-de-la-Rúa, Estanislao Gutierrez-Sanchez, Adoración Montero-Sánchez, Helena Lucena-Padros, Helder André, Francisco J. Díaz-Corrales

**Affiliations:** 1Department of Integrative Pathophysiology and Therapies, Andalusian Molecular Biology and Regenerative Medicine Centre (CABIMER), Junta de Andalucía, CSIC, University of Seville, University Pablo de Olavide, 41092 Seville, Spain; alvaro.plaza@cabimer.es (A.P.R.); dori.mntro@gmail.com (A.M.-S.); 2Department of Clinical Neuroscience, Division of Eye and Vision, St. Erik Eye Hospital, Karolinska Institutet, 17177 Stockholm, Sweden; filippo.locri@ki.se (F.L.); flavia.plastino@ki.se (F.P.); anders.kvanta@ki.se (A.K.); helder76andre@gmail.com (H.A.); 3Department of Ophthalmology, Virgen Macarena University Hospital, 41009 Seville, Spain; dramorillosanchez@gmail.com (M.J.M.-S.); erdelarua@gmail.com (E.R.-d.-l.-R.); esgusan@hotmail.com (E.G.-S.); 4Department of Surgery, Ophthalmology Area, University of Seville, 41009 Seville, Spain; 5Department of Maternofetal Medicine, Genetics and Reproduction, Institute of Biomedicine of Seville, CSIC, Virgen del Rocio University Hospital, University of Seville, 41013 Seville, Spain; drahelenalucenapadros@gmail.com

**Keywords:** age-related macular degeneration, vascular endothelial growth factor, biomarker, tear fluid, choroidal neovascularization

## Abstract

Vascular endothelial growth factor (VEGF) is a key mediator of exudative age-related macular degeneration (eAMD), yet non-invasive biomarkers for disease monitoring remain limited. This study evaluates VEGF levels in human tear fluid as a potential biomarker for eAMD and investigates the molecular dynamics of VEGF in a laser-induced choroidal neovascularization (lCNV) mouse model. Tear VEGF levels were quantified using proximity qPCR immunoassays in eAMD patients (n = 29) and healthy controls (n = 21) and correlated with optical coherence tomography (OCT) findings. Molecular analyses, including immunohistochemistry, gene expression profiling, and phosphorylation assays, were conducted on choroid–retinal pigment epithelium (RPE) and lacrimal gland (LG) tissues from lCNV mice (n = 25). Tear VEGF levels were significantly elevated in eAMD patients, correlating with disease severity. Females exhibited higher VEGF levels, a pattern not replicated in the mouse model. In lCNV mice, VEGF overexpression originated from the choroid–RPE, driven by hypoxic and inflammatory signaling, with no significant LG contribution. Increased VEGF, IL-6, and vimentin expression, along with NF-κB and STAT3 activation, were observed. These findings suggest that tear VEGF is a promising non-invasive biomarker for eAMD, warranting further validation for clinical application in disease monitoring and treatment optimization.

## 1. Introduction

Age-related macular degeneration is a debilitating condition and the leading cause of non-hereditary blindness among adults in developed countries [1]. AMD presents in two primary forms: non-exudative (dry) AMD and exudative (wet) AMD [2,3]. Early non-exudative AMD, accounting for 80–90% of cases, is characterized by the deposits of lipids and proteins that accumulate under the retinal pigment epithelium (RPE) and the Bruch’s membrane, known as drusen. The advanced form of non-exudative AMD is characterized by the degeneration of the RPE and cones, resulting in extensive macular atrophy known as geographic atrophy [3]. The prevalence of advanced non-exudative AMD is more than half of that of exudative age-related macular degeneration (eAMD). Conversely, eAMD, though less common, is more severe. This form is marked by choroidal neovascularization (CNV), where aberrant blood vessels proliferate, infiltrating the retina and causing edema and hemorrhages, potentially leading to central blindness. Notably, non-exudative AMD can progress to the exudative form, although some patients present with eAMD at onset [3].

Treatment options differ between AMD types. Geographic atrophy has few effective therapeutic options, mainly based on complement C3 and C5 inhibitors (Pegcetacoplan and Avacincaptad pegol), while eAMD is primarily managed with intravitreal injections of anti-vascular endothelial growth factor (VEGF) agents such as bevacizumab, ranibizumab, aflibercept, brolucizumab, and faricimab [4,5,6]. These therapies reduce VEGF bioavailability, a key factor in disease progression that promotes endothelial cell proliferation, inflammation, and neovascularization [4,7,8]. Anti-VEGF injections have significantly improved outcomes in eAMD by targeting CNV, yet their repeated administration poses risks, including endophthalmitis, increased intraocular pressure, and retinal detachment. The likelihood of such adverse events increases with the cumulative number of doses administered [9]. Therefore, accurately timing these injections, ideally before VEGF levels rise significantly, could reduce these risks.

Currently, the frequency of anti-VEGF injections is determined by clinical assessments of disease activity, highlighting the need for reliable biomarkers to guide treatment intervals [4,5,10]. An ideal biomarker would not only predict VEGF surges after therapy, but also enable early intervention to prevent disease progression. VEGF, secreted by retinal cells such as RPE, ganglion cells, Müller cells, pericytes, and endothelial cells, remains a pivotal factor in AMD [7,8,11]. While RPE cells release minimal VEGF under normal conditions, levels rise dramatically in pathological states [7,8,11].

Although optical coherence tomography (OCT) is the most helpful imaging tool for monitoring eAMD patients and determining when to administer anti-VEGF injections, there are currently no molecular biomarkers available to guide treatment decisions, predict disease progression based on the detection of pathological VEGF elevations, or forecast the transition from non-exudative to eAMD, leaving a critical gap in clinical management [12]. An improved method to determine when re-administration of anti-VEGF injections is required to enhance treatment efficacy and avoid unnecessary doses [12,13]. Additionally, a biomarker capable of predicting the transition from non-exudative to eAMD would be invaluable, potentially allowing for earlier intervention with anti-VEGF therapy and a more optimistic outlook for eAMD treatment [8]. Early intervention may not only minimize complications, but also improve treatment outcomes by curbing the progression of CNV and preventing severe vision loss associated with elevated VEGF levels [9].

This pilot study investigates VEGF levels in tears as a potential biomarker for eAMD. We hypothesize that VEGF can be detected in tears and that its levels are elevated in patients with eAMD. Additionally, we propose that tear VEGF levels correlate with disease activity and may differ between sexes. Finally, we aim to determine whether VEGF present in tears originates from lacrimal gland (LG) production or is secreted from the retina–RPE–choroid complex in mice. Using a mouse model of laser-induced CNV (lCNV), we examined VEGF levels in the RPE–choroid complex, intraocular fluid (IOF), and LG. By integrating findings from human samples and animal models, this study seeks to elucidate the origin of tear VEGF and its clinical relevance in eAMD pathogenesis, paving the way for novel, non-invasive diagnostic and therapeutic strategies.

## 2. Results

### 2.1. Demographic Data

A total of 50 participants were enrolled in this pilot study, comprising 29 patients with eAMD and 21 healthy controls. Among the eAMD patients, 18 were diagnosed with active eAMD, while 11 had terminal eAMD. The demographic characteristics and AMD-associated risk factors of the study population are summarized in Table 1. The mean age of all participants was 73.5 ± 10.7 years, with a significant difference between groups (*p* < 0.001). The control group had the lowest mean age (68.6 ± 10.3 years), while the active eAMD group had a mean age of 74.1 ± 9.2 years, and the terminal eAMD group had the highest mean age (81.9 ± 8.3 years). Terminal eAMD was more prevalent in male patients (63%), while active eAMD was equally distributed between males and females, with no significant differences in sex distribution between groups (*p* = 0.38). Notably, 17 participants (34%) were active smokers, 12 of whom were in the eAMD group (8 with active eAMD and 4 with terminal eAMD). Additionally, 19 participants (38%) were classified as overweight or obese, with 14 in the eAMD group (7 in each subgroup). A family history of AMD was reported in 14 participants (48%), with 9 belonging to the active eAMD group. HTN was the most common comorbidity, present in 55% of patients with active eAMD, 63% of those with terminal eAMD, and 24% of the control group. A significant association was identified between eAMD and HTN (*p* < 0.05). Cataract surgery was reported in 50% of patients with active eAMD, 55% of those with terminal eAMD, and 19% of the control group (*p* = 0.06). IHD was documented in 22% of the active eAMD group, 18% of the terminal eAMD group, and 9% of the control group. However, no statistically significant association was found between AMD and IHD.

### 2.2. VEGF Levels in Human Tears

The median levels of VEGF in tears were significantly higher in patients with eAMD compared to the controls, with median levels of 124.9 pg/mL (95% CI: 115.3–255.9) versus 36.46 pg/mL (95% CI: 26.53–50.46), respectively (*p* < 0.001). Among male participants, the median VEGF level in the eAMD group was 101.2 pg/mL (95% CI: 66.58–189.6), compared to 40.06 pg/mL (95% CI: 16.35–72.6) in the controls (*p* < 0.05). For female participants, the difference was even more pronounced, with the eAMD group exhibiting a median VEGF level of 166 pg/mL (95% CI: 116.9–395.8) versus 28.78 pg/mL (95% CI: 21.97–47.62) in the female controls (*p* < 0.001) (Table 2).

A significant difference was observed between female and male patients with eAMD (median difference: 85.67 pg/mL, 95% CI: 0.7163–284.5; *p* < 0.05). Conversely, no significant difference in VEGF levels was found between male and female controls (median difference: 11.29 pg/mL, 95% CI: −14.30–37.80; *p* = 0.64). VEGF levels were also independent of visual acuity (VA); patients with eAMD and VA ≥ 0.4 had similar VEGF levels to those with VA < 0.4 (median difference: 12.27 pg/mL, 95% CI: −101.2–120.2; *p* = 0.74). Central foveal thickness (CFT) did not show a significant association with VEGF levels (median difference: 7.61 pg/mL, 95% CI: −52.95–126.9; *p* = 0.35). Furthermore, VEGF levels did not differ significantly between patients receiving fewer than 10 intravitreal injections and those receiving 10 or more injections (median difference: 45.65 pg/mL, 95% CI: −50.76–277.6; *p* = 0.45) (Table 3).

The distribution of VEGF levels among all participants is illustrated (Figure 1A), showing a clear curve separation between the eAMD and control groups. The data confirm that median VEGF levels are significantly elevated in eAMD patients compared to controls (Figure 1A’; *p* < 0.001), consistent with the findings in Table 2. In an the age-matched subgroup analysis of eAMD patients and controls aged 80–85 years (n = 5 per group), VEGF levels remained significantly elevated in eAMD (mean difference: 164.3 pg/mL, 95% CI: 46.69–281.9; *p* < 0.01), suggesting the observed effect is independent of age. The further categorization of patients into active eAMD, terminal eAMD, and controls reveals distinct VEGF level distributions (Figure 1B). Significant increases in VEGF levels were observed in the active eAMD group compared to the controls (Figure 1B’; mean difference: 193.6 pg/mL, 95% CI: 91.42–295.7; *p* < 0.001). No significant difference was noted between the terminal eAMD and control groups (Figure 1B’; mean difference: 44.88 pg/mL, 95% CI: −77.30–167.0; *p* = 0.56). Additionally, a significant difference was found between the terminal eAMD and active eAMD groups (Figure 1B’; mean difference: 148.7 pg/mL, 95% CI: 23.27–274.1; *p* < 0.05). These findings suggest that VEGF levels are significantly elevated in active eAMD relative to both controls and terminal eAMD, highlighting its potential as a biomarker for differentiating stages of AMD (Figure 1B’).

Additionally, an analysis based on three subgroupings—number of involved eyes, sex, and OCT examination findings—provides further insights (Figure 1C–E). The analysis by the number of involved eyes reveals significant differences between the control and eAMD groups (Figure 1C). VEGF levels were lower in the “Both Eyes Control” group compared to both the “Both Eyes eAMD” (mean difference: 111.0 pg/mL, 95% CI: 10.85–211.2; *p* < 0.05) and “One Eye eAMD” groups (mean difference: 198.5 pg/mL, 95% CI: 37.51–359.4; *p* < 0.05). No significant differences were observed between the “Both Eyes eAMD” and “One Eye eAMD” groups (mean difference: −87.42 pg/mL, 95% CI: −244.5–69.68; *p* = 0.34), or between the “Fellow Eye of eAMD” and other groups (Figure 1C).

Sex-based analysis showed that female patients with eAMD had significantly higher VEGF levels compared to male patients (Figure 1D; Mann–Whitney test, *p* < 0.05; median difference: 85.67 pg/mL, 95% CI: 0.7163–284.5), aligning with the findings in Table 3. Lastly, VEGF levels were significantly higher in eAMD patients with intraretinal fluid and/or subretinal fluid compared to those without these OCT features (Figure 1E; *p* < 0.05; median difference: 127.3 pg/mL ± 60.30 SEM, 95% CI: 2.857–251.8).

### 2.3. VEGF Levels in lCNV

Given the observed elevation of VEGF levels in eAMD patients, we investigated VEGF protein expression in lCNV. Our study focused on the LG, choroid–RPE, and IOF. Western blot analysis revealed a significant increase in VEGF levels in both the LG (*p* < 0.05) and choroid–RPE (*p* < 0.0001) of lCNV mice compared to the controls (Figure 2A,A’). This suggests enhanced local VEGF production in response to laser-induced injury to choroid–RPE. The proximity qPCR immunoassay studies confirmed a significant rise in VEGF levels in the LG (mean difference: 327.5 pg/mL, 95% CI: 1.266–653.6; *p* < 0.05) and IOF (mean difference: 16.43 pg/mL, 95% CI: 5.738–27.12; *p* < 0.01) of lCNV mice compared to the controls (Figure 2B). No significant increase was observed in the choroid–RPE (mean difference: 3.457 pg/mL, 95% CI: −79.41–86.33; *p* = 0.93) by proximity qPCR immunoassays, likely due to VEGF secretion into the IOF (Figure 2B). This secretion may explain the elevated VEGF levels in the IOF, suggesting that the choroid–RPE is a potential source of VEGF contributing to tear fluid.

In the eAMD patient cohort, sex-based analysis revealed that female patients had significantly higher VEGF levels compared to male patients. To investigate this further, we performed a sex-based analysis in the LG, IOF, and choroid–RPE of the lCNV mouse model using proximity qPCR immunoassays. The analysis yielded *p*-values of 0.16, 0.35, and 0.19, respectively, indicating no significant differences in VEGF levels between male and female mice in any of these tissues (Figure 2C).

### 2.4. Transcription Factor Activation in Response to VEGF Elevation in the lCNV Mouse Model

Following the observed increase in VEGF levels across the LG, IOF, and choroid–RPE in the lCNV mouse model, we investigated the molecular mechanisms driving this elevation. To identify the source of VEGF upregulation, we analyzed changes in key transcription factors using Western blot analysis. As shown in Figure 3, there were no significant changes in HIF-1α levels in the LG of lCNV mice compared to the controls (Figure 3A’; *p* = 0.2972). However, a significant increase in HIF-1α was observed in the choroid–RPE (Figure 3A’; *p* < 0.05), indicating localized hypoxic stress in this tissue (Figure 3A). Laser-induced CNV also led to an enhanced phosphorylation of NF-κB p65 at Ser276 (Figure 3B) and STAT3 at Tyr705 (Figure 3C), both in the LG (Figure 3B’,C’; *p* < 0.05 and *p* < 0.05, respectively) and the choroid–RPE (Figure 3B’,C’; *p* < 0.01 and *p* < 0.001, respectively) when compared to the controls. Notably, this increase in phosphorylation was more pronounced in the choroid–RPE, suggesting a stronger inflammatory and signaling response in this tissue. The more pronounced response in the choroid–RPE highlights this tissue as a key contributor to VEGF upregulation and inflammation in the lCNV model, potentially driving the pathological neovascularization process.

### 2.5. Dynamic Transcriptional Regulation of Key Genes in the lCNV Model

To evaluate the effects of laser-induced CNV on the expression of VEGF, COX2, IL-6, and vimentin, we quantified mRNA levels in both the choroid–RPE and the LG. These genes are known to play crucial roles in the inflammatory and angiogenic processes associated with CNV [14,15,16]. As depicted in Figure 4, laser treatment resulted in a significant upregulation of IL-6, VEGF, and vimentin mRNA in the choroid–RPE. Specifically, IL-6 levels increased significantly post-laser treatment (Figure 4A; *p* < 0.01), indicating a strong inflammatory response in the choroidal tissue. VEGF demonstrated a highly significant elevation (Figure 4C; *p* < 0.001), reinforcing its role in CNV pathophysiology. Vimentin also increased significantly (Figure 4D; *p* < 0.05), suggesting its involvement in tissue structural changes during CNV progression.

In contrast, no significant changes in mRNA levels were observed for IL-6, VEGF, or vimentin in the LG, indicating that the LG does not contribute significantly to the local inflammatory or angiogenic processes in the lCNV model (Figure 4E–G). Additionally, COX2 mRNA levels remained unchanged in both the choroid–RPE and LG, indicating that the COX2 pathway is not activated in response to laser-induced CNV (Figure 4B,F).

### 2.6. Evaluation of LG Neovascularization and Immune Cell Recruitment in Response to VEGF

To evaluate VEGF-driven neovascularization and the recruitment of immune cells in the LG, we performed immunostaining for CD31, a marker of endothelial cells, and CD206, a marker for macrophages and dendritic cells. LG samples from the lCNV mice were compared to the controls. Additionally, an ex vivo experiment was conducted, wherein LG tissues from control mice were cultured with increasing concentrations of recombinant mouse VEGF (0, 50, 100, and 200 ng/mL) (Figure 5).

Immunohistochemical analysis of the LG from lCNV mice (Figure 5C,D) revealed an increased expression of CD31 and CD206 compared to the controls (Figure 5A,B), indicating enhanced vascularization and immune cell recruitment. To quantify these changes, immunofluorescence staining for CD31 and CD206 was performed (Figure 5E,F). A significant increase in CD31 membrane-bound protein integrated density was observed (Figure 5H; *p* < 0.01), reflecting heightened endothelial cell activity and an increase in the number of vessels/leukocytes [17]. However, the rise in CD206 expression, a marker of immune cell recruitment, was not statistically significant (Figure 5G; *p* = 0.20), suggesting a limited immune response in the LG of lCNV mice.

In the ex vivo experiment, the immunofluorescence analysis of LG using DAPI and CD31 was employed (Figure 5I,J). A trend of increased vessel density (vessels/mm^2^) was observed with higher VEGF concentrations, achieving statistical significance at 100 ng/mL (*p* < 0.05) and 200 ng/mL (Figure 5K; *p* < 0.01). This result demonstrates VEGF’s potent role in inducing LG neovascularization in a dose-dependent manner.

These findings underscore VEGF’s crucial role in promoting neovascularization in the LG of the lCNV model, as evidenced by increased CD31 expression and vessel density. The relatively modest CD206 response suggests that while VEGF plays a significant role in vascular changes, its influence on immune cell recruitment in the LG may be limited or indirect. The ex vivo data further support the hypothesis that VEGF secreted by the choroid–RPE is a driving factor in LG neovascularization, solidifying VEGF as a key mediator in angiogenesis within the LG.

## 3. Discussion

Exudative AMD presents a significant clinical challenge due to its progressive nature and impact on vision [18]. The disease is a leading cause of vision loss worldwide, and its management is complicated by the variability in disease progression and response to treatments. Biomarkers that can detect early disease changes or predict responses to treatment are invaluable for improving patient outcomes and minimizing the risk of severe vision loss [19]. Accurate biomarkers can provide critical insights into disease activity and treatment efficacy, allowing for more personalized and timely interventions.

Management strategies for AMD, especially for eAMD, rely heavily on intravitreal anti-VEGF therapies to inhibit pathological neovascularization [20]. Anti-VEGF therapies have revolutionized the treatment of eAMD by targeting the underlying angiogenic processes. However, the current reliance on subjective clinical assessments to determine the frequency of injections underscores a significant limitation in treatment protocols [21]. Frequent intravitreal injections are required based on clinical evaluations of disease activity, which can be subjective and vary between clinicians [9,21]. This approach not only poses risks of adverse events, but also increases the burden on patients and healthcare systems. [9]. Therefore, the need for more precise biomarkers that can signal when treatment is required or when disease activity is changing is critical to optimizing treatment regimens and improving patient outcomes [22].

The significantly elevated VEGF levels in the tears of patients with eAMD found in this study suggest that tear VEGF could serve as a non-invasive biomarker for monitoring disease activity in eAMD. Current OCT-guided regimens primarily rely on the detection of retinal fluid, such as intraretinal and subretinal fluid, to assess disease progression and guide treatment decisions. However, OCT imaging is resource-intensive and inherently subjective, depending on clinician interpretation. Using a biomarker in tears to measure VEGF levels in patients with eAMD could provide a complementary approach to OCT, offering an additional objective measure for disease assessment. Tear collection is a straightforward, non-invasive, and quick procedure, minimizing patient discomfort compared to the repetitive and time-intensive nature of OCT. A tear-collected biomarker would allow more frequent and tailored monitoring of VEGF levels, aiding in the early detection of pathological increases and enhancing the ability to predict the progression from non-exudative to eAMD. This approach could improve treatment decisions, such as the timely administration of anti-VEGF injections, while reducing the risks of both under treatment and unnecessary dosing.

Our OCT findings demonstrate a significant correlation between elevated VEGF levels and the presence of intraretinal and/or subretinal fluid, hallmarks of active eAMD. Patients with these OCT features exhibited significantly higher VEGF levels than those without fluid, underscoring the pivotal role of VEGF in retinal fluid accumulation and disease progression. The combination of VEGF quantification with OCT imaging could enhance the accuracy of disease monitoring, allowing clinicians to detect disease exacerbations earlier and adjust treatment protocols proactively.

A distinctive aspect of our study is the observation of sex-based differences in VEGF levels among eAMD patients, with female patients exhibiting significantly higher levels than their male counterparts. While we observed elevated VEGF levels in female patients, it remains unclear whether this increase stems from higher systemic VEGF levels or local secretion. Previous studies have reported sex-related differences in serum VEGF levels [23], suggesting a potential biological basis for these variations. The higher VEGF levels in women may reflect a more pronounced inflammatory or angiogenic response [23], potentially driven by hormonal or genetic factors. Nevertheless, a key limitation of this pilot study is the small sample size, which prevents us from drawing definitive conclusions regarding these inter-gender differences. Future studies with larger sample sizes are needed to determine whether elevated VEGF levels in female patients could influence therapeutic responses or increase the risk of progression to eAMD.

Additionally, our analysis revealed a higher incidence of intraretinal and/or subretinal fluid in female patients, indicating either more severe disease manifestations or a more rapid disease progression. These findings are consistent with observations in other ocular diseases, where sex differences in disease expression and outcomes have been documented [24]. Interestingly, in our animal model, sex-based differences in VEGF levels were not observed, possibly due to the acute nature of laser-induced CNV, which may not fully capture long-term, sex-specific disease dynamics. Further research is warranted to elucidate the mechanisms underlying these differences and explore their clinical implications. Incorporating sex-specific considerations into treatment strategies may help optimize outcomes for female patients with eAMD.

AMD primarily affects older individuals, many of whom face multiple comorbidities that can complicate direct clinical studies [25]. Animal models are therefore essential for assessing the validity and reliability of VEGF as a biomarker before clinical findings can be applied more broadly. To support our main study in humans, we performed experiments with lCNV. These experiments enabled us to examine VEGF expression in various ocular compartments and investigate how VEGF might potentially reach the LG, as well as its role in inflammation. This complementary approach provides important insights into eAMD pathophysiology and the broader systemic effects of VEGF, contributing to the development of biomarker-based strategies. Our findings aim to establish a foundation for more accurate and individualized eAMD treatments, improving patient outcomes while avoiding unnecessary interventions.

In our study, we found that VEGF levels were significantly elevated in the tears of patients with eAMD compared to controls. This elevation was consistently reproduced in the lCNV mouse model. Importantly, our molecular analyses indicated that the primary source of tear VEGF is the choroid–RPE complex, rather than the LG. This conclusion is supported by the absence of significant changes in HIF-1α expression in the LG, while a significant increase in HIF-1α was observed in the choroid–RPE, consistent with localized hypoxic stress driving VEGF upregulation. Furthermore, we observed an increased phosphorylation of NF-κB and STAT3 in the choroid–RPE, indicating the activation of inflammatory and stress-response pathways associated with VEGF production. These findings reinforce the interpretation that VEGF secreted in response to CNV originates from the posterior segment of the eye and may reach the tear film via intraocular fluid dynamics or systemic diffusion.

Gene expression analysis provided additional evidence, showing a significant upregulation of VEGF, IL-6, and vimentin in the choroid–RPE, consistent with the angiogenic and inflammatory processes characteristic of CNV. In contrast, no significant changes in these markers were observed in the LG, further supporting that the LG is not a major contributor to VEGF production in this context. Furthermore, immunohistochemical analysis demonstrated VEGF-induced neovascularization in the LG, evidenced by increased CD31 staining, though immune cell recruitment, as indicated by CD206 labeling, remained modest. This suggests that while VEGF promotes neovascularization in the LG, its impact on immune cell recruitment is limited. While the lCNV model recapitulates the key features of neovascular AMD, it does not fully reflect the chronic and multifactorial nature of the human disease. The observed changes in the LG should be interpreted as indirect effects of VEGF elevation, rather than as a direct model of AMD-associated lacrimal involvement. Collectively, these findings highlight the choroid–RPE as the primary source of VEGF in tears, positioning tear VEGF as a valuable biomarker for monitoring eAMD.

Tear samples from human subjects were analyzed using a proximity qPCR immunoassay. Our approach using this technique offers several advantages over traditional methods. Proximity qPCR immunoassays achieve high sensitivity by leveraging DNA amplification, enabling the detection of proteins at very low concentrations. Its ability to work effectively with minimal sample volumes makes it particularly suitable for detecting low-abundance biomarkers such as VEGF. Additionally, the dual-antibody recognition system in proximity qPCR immunoassays ensures high specificity, reducing the risk of cross-reactivity and false positives observed in some traditional immunoassay approaches [25,26]. These technical advantages allow for a more accurate and reliable measurement of VEGF levels in tear samples, improving our ability to assess their clinical relevance in eAMD.

Despite the promising findings, this study has several limitations. Although tear samples were collected using a standardized protocol, variability in tear composition due to environmental factors, reflex tearing, and individual differences remains a known limitation in tear biomarker research. The sample size was relatively small, which does not allow for the further stratification of patients with early and intermediate AMD as a control group. Including these stages of the disease would provide a more comprehensive perspective on the potential role of tear VEGF levels as a biomarker for disease progression. Additionally, the control group was not fully age-matched with the eAMD patients, which could have influenced VEGF levels, as aging is associated with changes in ocular and systemic vascular biology. However, the focused subgroup analysis of this study using a small sample of age-matched individuals (80–85 years) confirmed that VEGF levels remained significantly elevated in the eAMD group, supporting the robustness of our findings.

As a cross-sectional pilot study, our research did not evaluate longitudinal changes in tear VEGF following anti-VEGF therapy. The primary objective of this study was to explore the feasibility of tear VEGF as a potential biomarker for eAMD. Our findings pave the way for future studies with a larger cohort, enabling us to validate VEGF levels in tears not only in the context of disease progression, but also in response to anti-VEGF treatment. Notably, previous studies, such as the work by Shahidatul-Adha et al. (2022) [27], have also reported increased VEGF levels in tear fluid in AMD, further supporting the role of tear VEGF as a potential biomarker for eAMD. Future studies incorporating serial sampling across treatment time points will be critical to assess the dynamics of tear VEGF and its potential as a predictive or monitoring biomarker.

## 4. Materials and Methods

### 4.1. Recruitment of Study Participants and Tear Fluid Collection

All patients provided informed consent, and the study was conducted in accordance with ethical standards, approved by the ethical committee (ethical permit PEIBA 0728-N-19, approved by the Andalusian Biomedical Research Ethics Portal (Portal de Ética de la Investigación Biomédica de Andalucía, PEIBA) on 22 May 2019). Tear samples were collected in the morning between 11:00 and 14:00 at the Department of Ophthalmology, Virgen Macarena University Hospital. Patients diagnosed with eAMD were recruited, and demographic data—including family history of AMD, smoking status, medication history, history of cataract surgery, body mass index (BMI), ischemic heart disease (IHD), hypertension (HTN), and other systemic illnesses—were obtained through direct questioning and verified against medical records [28]. A comprehensive ophthalmic assessment was performed, including best-corrected visual acuity (BCVA), slit-lamp examination, dilated fundus examination, and OCT. All participants, including healthy controls, were screened for anterior segment diseases such as dry eye, conjunctivitis, and history of pterygium surgery. None of the included individuals presented with these conditions. Control subjects were recruited from patients with no retinal pathology or ocular surface disease. A full ophthalmic assessment was performed to confirm the absence of retinal disease in either eye. Due to the nature of their referral, control participants tended to be younger, which limited precise age matching. Patients were classified into active eAMD and terminal eAMD, following the Beckman Initiative classification [29]. Terminal eAMD was defined as cases where patients had been without treatment for at least three months, presented with macular fibrosis and/or atrophy, and had a BCVA of less than 0.05 [28]. Treatment was discontinued due to the absence of visual improvement in patients with terminal eAMD. Active eAMD patients were treated with 2 mg aflibercept following a “treat and extend” protocol, adjusting treatment intervals per clinical evolution in accordance with manufacturer guidelines. OCT imaging was performed using Swept Source OCT (SS-OCT) (DRI OCT TRITON; Topcon Corporation, Tokyo, Japan) with a 7 × 7 macular cube protocol, acquiring a single scan per eye at the time of tear sample collection. The analysis focused on detecting intraretinal fluid, subretinal fluid, pigment epithelium detachment, and the type of choroidal neovascular membrane (type 1, 2, or 3) [29,30]. Tears were collected using diagnostic ophthalmic filter paper (Schirmer strips). Each strip was inserted into the outer third of the lower eyelid and allowed to absorb tears for a minimum of 5 min, ensuring a wetting length of at least 10 mm. The strips were then carefully removed and placed in individual clear 0.5 mL polypropylene tubes. All tubes containing tear samples were immediately stored at −80 °C until further analysis.

### 4.2. Proximity Real-Time PCR Immunoassays

VEGF-A protein levels were quantified using the Human/Mouse VEGF ProQuantum Immunoassay Kit (Thermo Fisher Scientific, Waltham, MA, USA) according to the manufacturer’s instructions. This assay utilizes pairs of oligonucleotide-labeled antibody probes specific to VEGF-A. When these probes bind in close proximity to the target protein, a PCR target sequence is generated through a proximity-dependent DNA polymerization event. The resulting PCR products were quantified using the CFX Connect Real-Time PCR detection system and CFX Manager software 3.0 (Bio-Rad Laboratories, Hercules, CA, USA), ensuring high sensitivity and specificity in protein quantification [31]. Human tear samples and mouse IOF samples were directly assessed using the assay kit. Due to the complex nature of LG samples, an additional step of chloroform/methanol precipitation of proteins was implemented to enhance protein purity. Experimental procedures were conducted in duplicate to ensure reproducibility and accuracy. Protein levels were expressed as picograms per milliliter (pg/mL) of the sample.

### 4.3. Animals and CNV Induction

This study adhered to the ARVO statement for the use of animals in ophthalmologic and vision research, and all protocols were approved by Stockholm’s committee for ethical animal research (ethical permit Dnr 7053-2020, approved by the Stockholm Animal Ethics Committee (Stockholms djurförsöksetiska nämnd) on 21 April 2020). Twenty-five C57BL/6J mice (Charles River, Cologne, Germany) were housed under a 12 h light/dark cycle with ad libitum access to food and water and were monitored daily for health and welfare. CNV lesions were induced in 8-week-old mice using a laser-coupled Micron IV system (Phoenix Research Labs, Pleasanton, CA, USA) with parameters set at 50 μm spot size, 180 mW intensity, and 100 ms duration, following established protocols [32,33]. Ten mice underwent the induction of four CNV lesions per eye, while an additional ten age-matched untreated control mice, consisting of both males and females, were included in the study. The remaining mice were allocated for ex vivo experiments. Both the experimental and control groups were deeply anesthetized and euthanized by cervical dislocation. Eyeballs and LGs were immediately excised (eyes dissected to obtain RPE–choroid complexes), frozen in liquid nitrogen, and stored at −80 °C until further analysis.

### 4.4. Animal Tears, Aqueous and Vitreous Humors, and RPE–Choroid Complex Dissection

The rodent model of CNV was compared with age-matched controls. On day 9 post-CNV induction, mice were euthanized in accordance with ethical guidelines. Prior to dissection, aqueous and vitreous humors were collected by extravasation of the IOF due to the limited anatomical size of the rodents’ eyes. Extraocular LGs and eyes were carefully enucleated, with the surrounding tissues meticulously removed. Moreover, for each experimental condition, RPE–choroid complexes were isolated from the neuroretina. Additionally, a small section of each LG was excised, post-fixed overnight in 4% paraformaldehyde in 0.1 M phosphate buffer (PB), and subsequently processed for paraffin embedding for immunohistochemistry, while the remainder was frozen in liquid nitrogen for molecular analysis, as performed with IOF and RPE–choroid complexes [34,35]. These samples underwent comprehensive evaluation using quantitative real-time PCR (qPCR), Western blotting, and proximity qPCR immunoassays.

### 4.5. Ex Vivo Culture and Treatment

Prior to culturing, glands were rinsed for 5 min in PB saline (PBS) containing 1% penicillin–streptomycin (P/S; Thermo Fisher Scientific, Waltham, MA, USA; cat no 15140122) and 2 μg/mL amphotericin B (Thermo Fisher Cat 15290026). On the first day, the LGs were cultured for 24 h in DMEM/F12 medium (a 1:1 mixture of Dulbecco’s Modified Essential Medium and Ham’s F-12 Medium; ThermoFisher Scientific Inc., Waltham, MA, USA) supplemented with 1% P/S and 10% fetal bovine serum (FBS; ThermoFisher Scientific Inc. A5256701). On the second day, the glands were transferred to a fresh medium with the same essential supplements, along with 0, 50, 100, or 200 ng/mL of recombinant human VEGF 165 protein (R&D Systems; cat no 293-VE-010/CF) for 48 h. The LGs were maintained in a standard cell culture incubator (20% oxygen, 5% carbon dioxide at 37 °C). On the fourth day, the glands were rinsed with PBS, fixed for 24 h, and then processed for paraffin embedding.

### 4.6. Tissue Processing for Histology and Immunohistochemistry

Paraffin-embedded LGs from six mice (totaling 12 glands) were sectioned using a microtome. Sections were deparaffinized, with 4 µm thickness used for morphometric analysis and 6 µm for immunohistochemistry. The morphometric analysis of hematoxylin and eosin (H&E) stained sections was conducted using an Axioskop 40 microscope (Zeiss, Gottingen, Germany), with images acquired at three different depth levels of the LG using a VisiCam TC10 camera (VWR, Lutterworth, UK). For immunohistochemistry, deparaffinized sections underwent antigen retrieval in EDTA buffer (pH 9) at 100 °C for 20 min. Immunoreactions were performed using a Bond III robotic system (Leica Biosystems, Newcastle, UK), following previously established protocols [33]. The primary antibodies anti-mouse CD31 and CD206 (diluted 1:50 and 1:100, respectively, in 0.1 M PB with 0.5% Triton X-100), were applied (details in Appendix A). The secondary antibodies included donkey anti-mouse Alexa Fluor 647 conjugate (Thermo Fisher Scientific, Waltham, MA, USA; Cat. No. A31571 and A11010, respectively). Hoechst 33,258 (Sigma Aldrich Corp., St. Louis, MO, USA; Cat. No. 14530) was used for counterstaining. Sections were mounted with a fluorescence mounting medium (Dako, Carpinteria, CA, USA) and imaged using an Axioskop 2 plus fluorescence microscope with AxioVision software (Version 4.6, Zeiss, Gottingen, Germany).

### 4.7. Western Blot

Western blot experiments were performed using two RPE–choroid complexes or LG from one mouse for each experimental condition. Samples were sonicated in RIPA buffer (Abcam ab156034, Cambridge, UK) containing protease and phosphatase inhibitor cocktails, then centrifuged at 15,000× *g* for 20 min at 4 °C. The supernatants, containing cytosolic and interstitial proteins, were used to detect soluble proteins. Protein concentration was determined using a Detergent Compatible Bradford Assay Kit (Thermo Fisher, Waltham, MA, USA cat no 23246). Aliquots of each sample containing equal amounts of protein (30 μg) were subjected to SDS-PAGE, with β-actin used as the loading control. The gels were trans-blotted onto PVDF (polyvinylidene difluoride) membranes (Bio-Rad Laboratories), and the blots were blocked in 5% non-fat milk in Tris-buffered saline (TBS; 20 mM Tris, 500 mM NaCl, pH 7.4) for 1 h at room temperature. They were then incubated overnight at 4 °C with primary antibodies (details in Appendix A). Secondary antibodies (anti-rabbit-IgG cat. no. P044801-2 or anti-mouse-IgG cat. no. P044701-2, conjugated to horseradish peroxidase, 1:10,000; Dako) were applied for 1 h at room temperature and developed with an enhanced chemiluminescence reagent. Images were acquired using a ChemiDoc XRS+ system (Bio-Rad Laboratories, Inc.), and the optical density (OD) of the bands was evaluated with Image Lab 3.0 software (Bio-Rad Laboratories, Inc.). Data were normalized to the corresponding OD of β-actin, STAT3, or NF-κB, as appropriate. All experiments were performed in duplicate.

### 4.8. Quantitative PCR

qPCR experiments were performed using the CFX96 Touch Real-Time PCR Detection System (Bio-Rad), with each sample containing two LGs or two RPE/choroids from one mouse per experimental condition. Total RNA was extracted using the RNeasy Mini Kit (Qiagen, Valencia, CA, USA), purified, resuspended in RNase-free water, and quantified using a fluorometer (Invitrogen, Carlsbad, CA, USA). First-strand cDNA was generated from 1 µg of total RNA using the QuantiTect Reverse Transcription Kit (Qiagen, Valencia, CA, USA). Gene expression was evaluated using an SYBR Green PCR Kit (Qiagen, Inc.). qPCR amplification was performed with SsoAdvanced Universal SYBR Green Supermix (Bio-Rad Laboratories, Hercules, CA, USA) on a CFX Connect Real-Time PCR detection system, with data analyzed using CFX Manager software (Bio-Rad Laboratories, Hercules, CA, USA). qPCR primer sets for VEGF, markers of epithelial-to-mesenchymal transitions, and inflammatory markers were chosen to hybridize to unique regions of the appropriate gene sequence (Appendix A for a complete list of assayed genes and primers). Target genes were assayed concurrently with ribosomal protein L13a (RpL13a), a constitutively expressed gene encoding a ribosomal protein. Samples were compared using the relative Ct (threshold cycle) method. The fold change in gene expression was determined relative to control mice after normalization to RpL13a. All reactions were performed in triplicate.

### 4.9. Statistical Analysis

Data normality was assessed using the Shapiro–Wilk test. Statistical analyses were performed using GraphPad Prism 10 software (GraphPad Software, Inc., San Diego, CA, USA). Outliers were identified and removed using Tukey’s method. Depending on the data distribution, either an unpaired Student’s *t*-test, Mann–Whitney test, or one-way ANOVA followed by Tukey’s multiple comparisons test with a single pooled variance was used. Results are expressed as mean ± standard error of the mean (SEM) based on the indicated n values or median difference 95% confidence interval (CI), as appropriate. A *p*-value of <0.05 was considered statistically significant.

## 5. Conclusions

Our pilot study demonstrates that VEGF levels in tears are a promising non-invasive biomarker for monitoring the progression of eAMD. Using a highly sensitive proximity qPCR immunoassay, we found significantly elevated VEGF levels in eAMD patients compared to healthy controls, with the highest levels observed in active cases with macular edema and lower levels in advanced atrophic stages. These findings correlate with OCT-based clinical observations.

Additionally, we identified a sex-based difference, with higher VEGF levels in female patients, suggesting a potential avenue for personalized treatment approaches. In animal models, we confirmed that VEGF detected in the LG originates primarily from the choroid–RPE complex rather than from intrinsic LG production. This suggests a similar mechanism in humans, where tear VEGF likely reflects RPE-derived secretion rather than direct LG synthesis.

These results highlight the utility of proximity qPCR immunoassays in detecting low-abundance biomarkers and emphasize the importance of considering sex differences in eAMD management. Future multicenter studies are necessary to validate these findings and assess the broader applicability of tear VEGF as a biomarker for eAMD.

## Figures and Tables

**Figure 1 ijms-26-03855-f001:**
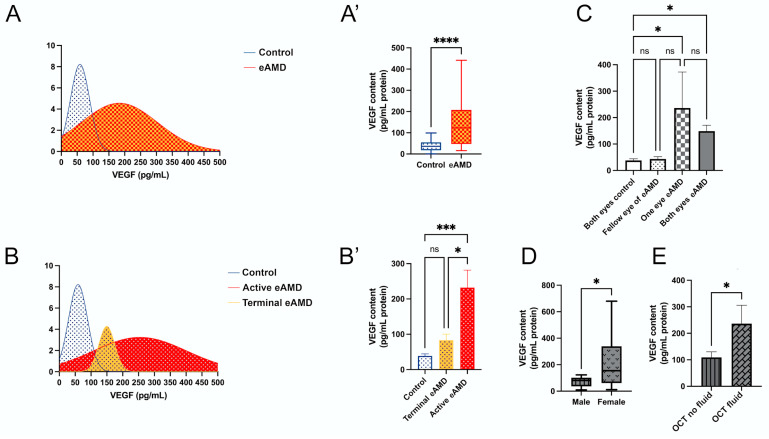
Distribution and analysis of VEGF levels in eAMD (eAMD) and control groups. (**A**,**A’**) VEGF level distribution in eAMD and control group. (**A**) Density plot showing the range of VEGF levels. (**A’**) Box plot comparing VEGF levels between eAMD and control groups. (**B**,**B’**) Stratification of VEGF levels across active eAMD, terminal eAMD, and control groups. (**B**) Density plot illustrating the distribution of VEGF levels among these subgroups. (**B’**) Bar plot comparing VEGF levels across the groups. (**C**) VEGF levels stratified by the number of affected eyes, with subgroups including both eyes as controls, both eyes affected in eAMD, one eye affected in eAMD, and fellow eye of eAMD-affected eye. (**D**) VEGF levels in male and female eAMD patients presented in a box plot. (**E**) VEGF levels in eAMD patients with or without intraretinal or subretinal fluid evaluated by OCT. Statistical analyses are noted based on data distribution, including Mann–Whitney test, one-way ANOVA with Tukey’s post hoc test, and unpaired Student’s *t*-test, with results expressed as mean ± SEM or median difference (95% CI), as applicable. * *p* < 0.05, *** *p* < 0.001, **** *p* < 0.0001.

**Figure 2 ijms-26-03855-f002:**
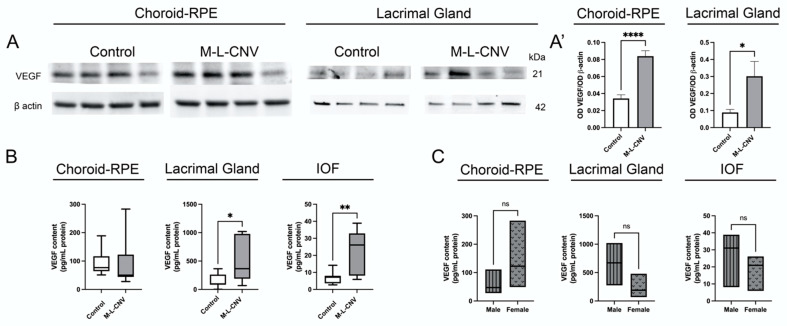
VEGF expression in lCNV mice. (**A**,**A’**) Western blot analysis of VEGF protein levels in the lacrimal gland (LG) and choroid–retinal pigment epithelium (choroid–RPE) of lCNV mice and controls. (**A**) Representative Western blot images showing VEGF levels in these tissues. (**A’**) Quantitative analysis of VEGF levels normalized to actin as a loading control. (**B**) Proximity ligation-based immunoassay (PLA) results showing VEGF levels in the LG, intraocular fluid (IOF), and choroid–RPE of lCNV mice. (**C**) VEGF levels in the LG, IOF, and choroid–RPE stratified by sex in lCNV mice, as assessed using PLA assays. Statistical analyses were performed based on data distribution using the Shapiro–Wilk test; unpaired Student’s *t*-test was used for (**A’**), while the Mann–Whitney test was used for B and C. Results are expressed as mean ± SEM or median difference (95% CI), as appropriate. * *p* < 0.05, ** *p* < 0.01, **** *p* < 0.0001.

**Figure 3 ijms-26-03855-f003:**
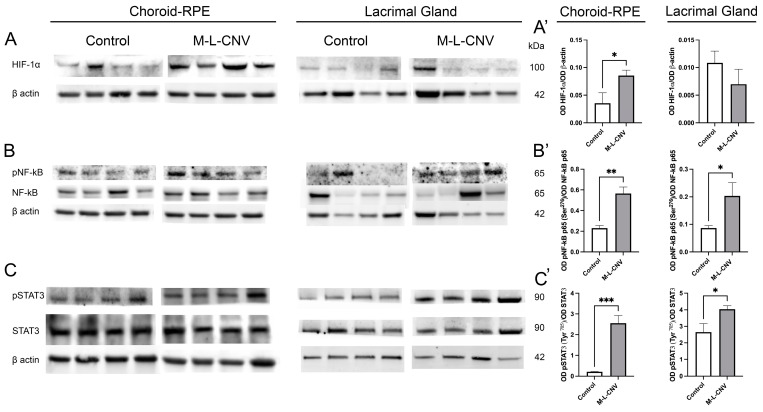
Western Blot analysis of transcription factor expression in choroid–RPE and lacrimal gland (LG) of lCNV and control mice. (**A**–**C**) Western blot analysis of HIF-1α, phosphorylated NF-κB p65 (Ser276), total NF-κB p65, phosphorylated STAT3 (Tyr705), and total STAT3 in the choroid–RPE and LG of lCNV and control mice. Actin served as the loading control. (**A’**) Densitometric analysis of HIF-1α levels in the choroid–RPE and LG. (**B’**,**C’**) Densitometric analysis of phosphorylated NF-κB p65 (Ser276) and phosphorylated STAT3 (Tyr705), normalized to total NF-κB p65 and total STAT3, respectively. Statistical analyses were performed using an unpaired Student’s *t*-test. Data are expressed as mean ± SEM from five independent samples (* *p* < 0.05, ** *p* < 0.01, *** *p* < 0.001).

**Figure 4 ijms-26-03855-f004:**
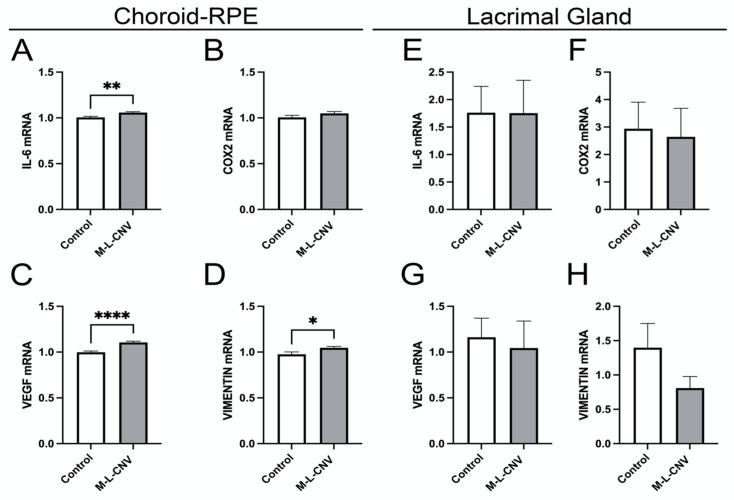
Dynamic transcriptional regulation of key genes in the lCNV model. (**A**–**D**) mRNA levels of IL-6, VEGF, COX2, and vimentin in the choroid–RPE of lCNV and control mice, assessed by qPCR. (**A**) IL-6 levels, (**B**) COX2 levels, (**C**) VEGF levels, (**D**) vimentin levels. (**E**–**H**) mRNA levels of IL-6, VEGF, COX2, and vimentin in the LG, assessed by qPCR. Data were analyzed by the 2^−ΔΔCT method with RpL13a as the internal standard. Statistical analysis was performed using an unpaired Student’s *t*-test. Results are presented as mean ± SEM from eight independent samples per group. (* *p* < 0.05, ** *p* < 0.01, **** *p* < 0.0001).

**Figure 5 ijms-26-03855-f005:**
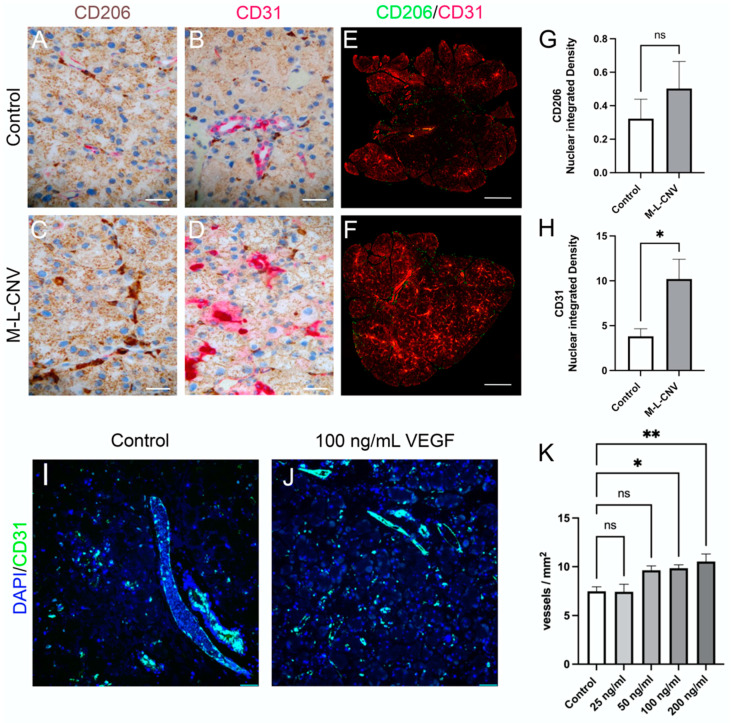
VEGF-mediated neovascularization and immune cell dynamics in the LG following lCNV. (**A**,**B**) Immunostaining for CD31 and CD206 in control LG samples. (**C**,**D**) Immunostaining for CD31 and CD206 in lCNV LG samples. (**E**,**F**) Quantitative immunofluorescence analysis of CD31 (**H**) and CD206 (**G**) expression. (**I**,**J**) Ex vivo LG cultures treated with recombinant VEGF (25, 50, 100, and 200 ng/mL). (**K**) Vessel density in response to VEGF treatment. Data are expressed as mean ± SEM and analyzed using one-way ANOVA or unpaired Student’s *t*-test (* *p* < 0.05, ** *p* < 0.01). Four independent samples were evaluated in each experiment.

**Table 1 ijms-26-03855-t001:** Demographic characteristics and systemic profiles of the subjects.

Variable	Total Subjects (n = 50)	Active eAMD (n = 18)	Terminal eAMD (n = 11)	Control (n = 21)	*p* Value
Age (years)					
(Mean ± SD)	73.5 ± 10.7	74.1 ± 9.2	81.9 ± 8.3	68.6 ± 10.3	<0.001 ^a^
Sex (n, %)					
Male	24 (48)	9 (50)	7 (63)	8 (38)	0.38 ^b^
Female	26 (52)	9 (50)	4 (37)	13 (62)	
Lifestyle (n, %)					
Cigarette smoking	17 (34)	8 (44)	4 (37)	5 (24)	0.39 ^b^
BMI (n, %)					
Overweight	10 (20)	3 (17)	3 (27)	4 (19)	0.44 ^c^
Obesity	9 (18)	4 (22)	4 (37)	1 (5)	
Family history of AMD (n, %)	20 (40)	9 (50)	5 (45)	6 (28)	0.36 ^b^
Comorbidity (n, %)					
HTN	22 (44)	10 (55)	7 (63)	5 (24)	0.04 ^b^
IHD	8 (16)	4 (22)	2 (18)	2 (9)	0.61 ^c^
Cataract surgery (n, %)	19 (38)	9 (50)	6 (55)	4 (19)	0.06 ^b^

AMD: age-related macular degeneration, BMI: body mass index, IHD: ischemic heart disease, HTN: hypertension, SD: standard deviation. Statistics: ^a^ One-way ANOVA. ^b^ Chi-square test. ^c^ Fisher’s exact test. *p* < 0.05 was considered significant.

**Table 2 ijms-26-03855-t002:** Comparison of median difference in VEGF levels between the groups.

Groups	Total		Male		Female	
VEGF	Control	eAMD	Control	eAMD	Control	eAMD
Median (pg/mL) (95%, CI)	36.46 (26.53–50.46)	124.9 *** (115.3–255.9)	40.06 (16.35–72.6)	101.2 * (66.58–189.6)	28.78 (21.97–47.62)	166 *** (116.9–395.8)

CI: confidence interval. Statistics: Mann–Whitney test. * *p* < 0.05; *** *p* < 0.001.

**Table 3 ijms-26-03855-t003:** Comparison of median difference in VEGF levels between subgroups.

Variable	Subgroups	Median Difference (pg/mL) (95%, CI)	*p*-Value
Sex	Female eAMD vs. Male eAMD	85.67 (0.7163–284.5)	0.04
	Male control vs. Female control	11.29 (−14.30–37.80)	0.64
VA of eAMD	VA ≥ 0.4 vs. VA < 0.4	12.27 (−101.2–120.2)	0.74
CFT of eAMD	CFT ≥ 250 vs. CFT < 250	7.61 (−52.95–126.9)	0.35
Intravitreal injection	Injection < 10 vs. Injection ≥ 10	45.65 (−50.76–277.6)	0.45

VA: visual acuity, CFT: central foveal thickness, CI: confidence interval. Statistics: Mann–Whitney test. *p* < 0.05 was considered significant.

## Data Availability

The raw data supporting the conclusions of this article will be made available by the authors on request.

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
