# Peer review of "VEGF in Tears as a Biomarker for Exudative Age-Related Macular Degeneration: Molecular Dynamics in a Mouse Model and Human Samples"

_ijms, 2025, doi:10.3390/ijms26083855_

Round 1
Reviewer 1 Report
Comments and Suggestions for Authors
The authors investigated the tear VEGF levels in AMD. The examination is non-invasive, and the result is interesting. Meanwhile, there are some questions about the study design and presentations. In addition, the discussion about the clinical relevance of the result seems to be an overstatement. Specific comments are listed below.
- Tear cytokine levels should be primarily affected by the anterior segment diseases. The authors should have checked dry eye, conjunctivitis, pterygium surgery, etc. If the data is not available, it should be discussed as a limitation.
- There is a significant concern for the interpretation of the animal model. Laser induced CNV is certainly a model of CNV; however, it is not the exact model of AMD. Analysis of CNV in comparison to AMD-related CNV is widely accepted. However, lacrimal grand after laser irradiation is not assumed a good model of lacrimal grand of AMD patients. Were the control animal underwent lasers that do not induce CNV? Focusing on the human data may emphasize the main purpose of the study.
- It is not clear how the control group were enrolled (Line 107). Actually, the age is not matched. How about using the fellow eyes as a control? If the VEGF is also elevated in the fellow eyes, it would be an interesting findings.
- Inclusion of early and intermediate AMD as a control would have make the result more interesting. The result may show the potential of tear VEGF level as a biomarker for detecting progression to neovascular AMD.
- The authors emphasized the inter-gender difference(line 540); however, the sample size is not very big to convince the difference. For example, female terminal AMD group consisted of only four patients. Discussion about disease activity may also a result of individual differences rather than inter-gender difference.
- The authors emphasized the effectiveness of tear VEGF levels as a biomarker compared to OCT(line 514). Although collecting tear fluid is certainly easy, the measurement of VEGF level is not as easy as the collection. The authors should be cautious about the claiming the superiority.
- In introduction(Line 63), the authors introduced C3 and C5 inhibitors as a treatment of non-exudative AMD. However, they are a treatment for geographic atrophy. The term non-exudative AMD may include early and intermediate AMD and is confusing.
Author Response
The authors investigated the tear VEGF levels in AMD. The examination is non-invasive, and the result is interesting. Meanwhile, there are some questions about the study design and presentations. In addition, the discussion about the clinical relevance of the result seems to be an overstatement. Specific comments are listed below.
We thank the reviewer for the constructive feedback and for recognizing the value of our study. We appreciate the insightful comments regarding study design, data presentation, and clinical interpretation, which have helped us improve the manuscript. Please find our detailed responses below.
1. Tear cytokine levels should be primarily affected by the anterior segment diseases. The authors should have checked dry eye, conjunctivitis, pterygium surgery, etc. If the data is not available, it should be discussed as a limitation.
We appreciate this important comment. All participants, including both eAMD patients and healthy controls, underwent slit-lamp examination to rule out anterior segment diseases such as dry eye, conjunctivitis, or history of pterygium surgery. None of the individuals included in the study presented with these conditions. This information has now been explicitly added to the manuscript (see revised text in Section2.1, Line 119-125).
2. There is a significant concern for the interpretation of the animal model. Laser induced CNV is certainly a model of CNV; however, it is not the exact model of AMD. Analysis of CNV in comparison to AMD-related CNV is widely accepted. However, lacrimal grand after laser irradiation is not assumed a good model of lacrimal grand of AMD patients. Were the control animal underwent lasers that do not induce CNV? Focusing on the human data may emphasize the main purpose of the study.
We thank the reviewer for raising a key point: it is due to the lack of association with laser induced CNV that we the authors hypothesized the experiments. To clarify, in our study, the lacrimal gland (LG) was not subjected to laser irradiation. Instead, we analyzed LG tissue from animals in which laser-induced CNV was applied to the posterior pole, targeting the retina and choroid. The is supportive that pathological VEGF produced in the retina/choroid (as demonstrated by the activation of transcription factors) during neovascularization could affect distant ocular tissues, such as the LG.
The rationale for using the laser-induced CNV (lCNV) model lies in its well-established role in mimicking the wound healing response and neovascularization associated with the late stages of human exudative AMD. It effectively models the VEGF-driven processes that characterize CNV in AMD.
Furthermore, to strengthen our findings and exclude direct laser effects, we conducted an ex vivo experiment using LG tissue from untreated wild-type mice, cultured with increasing concentrations of recombinant VEGF. The observed dose-dependent neovascularization in the LG confirms that the changes we observed in vivo are likely driven by systemic or local diffusion of VEGF secreted from the retina/choroid, rather than direct laser-induced damage to the LG.
Regarding controls: To date, there are no conclusive studies on the systemic effects of m-l-CNV, apart from a few indexed abstracts, which suggest there are no known systemic events post lCNV. Inducing non-CNV laser controls would risks of inducing other tissues’ aberrations, therefore putative uncharacterized models of disease that even from an ethical perspective would have considerable limitations. We opted top take the conventional approach for animal research and used: Ten age-matched untreated control mice, consisting of both males and females, were included in the study non-lasered eyes from the same animals served as internal controls, and untreated animals were also included as independent controls to ensure accurate comparison. (see revised text in Line 170).
Lastly, we agree that the human data are central to the translational value of the study, and in the revised manuscript we have emphasized the human findings more clearly in both the Discussion sections. This information has now been explicitly added to the manuscript (see revised text in Line 582-593 and 602-606).
3. It is not clear how the control group were enrolled (Line 107). Actually, the age is not matched. How about using the fellow eyes as a control? If the VEGF is also elevated in the fellow eyes, it would be an interesting finding.
We thank the reviewer for this insightful and important comment. We have clarified the recruitment strategy for the control group in the revised Methods section. Control participants were recruited from patients referred to the ophthalmology clinic for evaluation of refractory surface diseases. All controls underwent a full comprehensive ophthalmic assessment to confirm the absence of retinal or anterior segment pathology.
Regarding the suggestion of using fellow eyes as internal controls, we agree this would be valuable. However, in our cohort, the majority of eAMD patients had bilateral disease or signs of retinal involvement in the fellow eye, limiting the feasibility of using the contralateral eye as a control.
We also acknowledge the limitation in age-matching between groups. This was primarily due to the younger demographic of the refractory surface disease patients used as controls. As this study was designed as a pilot investigation to explore our hypothesis, we agree that a larger, age-matched cohort is necessary to validate and extend our findings.
Nonetheless, to address this concern, we performed a subgroup analysis comparing five age-matched controls (80–85 years) with five eAMD patients in the same age range. VEGF levels remained significantly higher in the eAMD group, supporting the robustness of the observed difference. (see revised text in Section2.1, Line 119-125 and Line 620-643 of discussion).
4. Inclusion of early and intermediate AMD as a control would have make the result more interesting. The result may show the potential of tear VEGF level as a biomarker for detecting progression to neovascular AMD.
We appreciate the reviewer’s suggestion regarding the inclusion of early and intermediate AMD as a control group. While we acknowledge that this would provide greater depth to the analysis, this pilot study was conducted with a limited sample size, which restricts the possibility of further stratification. Nevertheless, our findings lay the groundwork for future studies with a larger cohort, allowing us to validate tear VEGF as a potential biomarker for disease progression as well as for assessing changes before and after anti-VEGF treatment. We now include a brief comment on this limitation in the discussion. (see revised text in Line 620-643 of discussion section).
5. The authors emphasized the inter-gender difference (line 540); however, the sample size is not very big to convince the difference. For example, female terminal AMD group consisted of only four patients. Discussion about disease activity may also a result of individual differences rather than inter-gender difference.
We appreciate the reviewer’s comment and agree that the small sample size limits our ability to draw definitive conclusions regarding inter-gender differences in VEGF levels. We have acknowledged this limitation in the discussion and have revised the manuscript accordingly. (see revised text in Line 549-570).
6. The authors emphasized the effectiveness of tear VEGF levels as a biomarker compared to OCT(line 514). Although collecting tear fluid is certainly easy, the measurement of VEGF level is not as easy as the collection. The authors should be cautious about the claiming the superiority.
We appreciate the reviewer’s comment and acknowledge the need for a more balanced comparison. We have revised the manuscript accordingly, replacing the sentence in line 532-534 with the following:
…Using a biomarker in tears to measure VEGF levels in patients with eAMD could provide a complementary approach to OCT, offering an additional objective measure for disease assessment…
7. In introduction(Line 63), the authors introduced C3 and C5 inhibitors as a treatment of non-exudative AMD. However, they are a treatment for geographic atrophy. The term non-exudative AMD may include early and intermediate AMD and is confusing.
We appreciate the reviewer’s clarification and agree that the term 'non-exudative AMD' may be misleading in this context. We have revised the manuscript accordingly, replacing 'non-exudative AMD' with 'geographic atrophy' to accurately describe the condition for which C3 and C5 inhibitors are being investigated. (see revised text in Line 63).
Reviewer 2 Report
Comments and Suggestions for Authors
The study of Moshtaghion and collaborators evaluates VEGF levels in human tear fluid as a potential biomarker for exudative AMD (eAMD) and investigates its molecular dynamics in the laser-induced choroidal neovascularization (lCNV) mouse model. This is a very nice study supporting the potential of VEGF levels in tear fluid as a biomarker for eAMD that could eventually help clinicians in their decision-making for intravitreal injections of anti-VEGF for eAMD treatment.
Authors show that VEGF is increased in patients with active eAMD compared to late stage eAMD and healthy subjects. If VEGF levels in tears were to be used as biomarkers for decision-making, it would be valuable to know VEGF levels in samples from eAMD patients undergoing anti-VEGF treatment at different time points to evaluate VEGF changes with time and its correlation with treatment. It is true that authors show a correlation with VEGF levels and the presence of intrarretinal fluid and other OCT parameters, but a timeline of VEGF in patients with eAMD would greatly help in the understanding of tear-VEGF as a biomarker for decision-making.
The main limitation of tears as a source for biomarkers is the high variability in composition due to multiple factors including time and method of tear collection. Therefore, authors should at least mention such limitation or if possible, provide VEGF levels of the same subjects at different sampling points.
Authors should mention other studies that measured VEGF in tear fluid in AMD and discuss their results. Unlike the present study, a previous work (https://www.nature.com/articles/s41598-022-08492-7) found VEGF in tear fluid increased in late AMD.
The in vivo work with the lCNV mouse model elegantly supports the rationale of VEGF changes as a result of VEGF changes in the RPE due to neovascularization/injury.
Minor:
-Introduction: please define eAMD
-Title 2.1 is unclear
-Line 179 “would” would be more appropriate than “will”
-Line 557 “Animals” should be in lower case, “animals”
Author Response
Comments and Suggestions for Authors
The study of Moshtaghion and collaborators evaluates VEGF levels in human tear fluid as a potential biomarker for exudative AMD (eAMD) and investigates its molecular dynamics in the laser-induced choroidal neovascularization (lCNV) mouse model. This is a very nice study supporting the potential of VEGF levels in tear fluid as a biomarker for eAMD that could eventually help clinicians in their decision-making for intravitreal injections of anti-VEGF for eAMD treatment.
- Authors show that VEGF is increased in patients with active eAMD compared to late stage eAMD and healthy subjects. If VEGF levels in tears were to be used as biomarkers for decision-making, it would be valuable to know VEGF levels in samples from eAMD patients undergoing anti-VEGF treatment at different time points to evaluate VEGF changes with time and its correlation with treatment. It is true that authors show a correlation with VEGF levels and the presence of intrarretinal fluid and other OCT parameters, but a timeline of VEGF in patients with eAMD would greatly help in the understanding of tear-VEGF as a biomarker for decision-making.
We sincerely thank the reviewer for the positive and encouraging feedback on our study. We appreciate the insightful comment regarding the value of assessing VEGF levels longitudinally in patients undergoing anti-VEGF therapy. As this was a pilot study aimed at evaluating the initial hypothesis and the feasibility of detecting tear VEGF as a biomarker, our current dataset does not include multiple time points from the same patients. However, we fully agree that a longitudinal assessment would significantly enhance the clinical applicability of this approach, and we are currently planning a larger-scale study with serial sampling to address this point.
- The main limitation of tears as a source for biomarkers is the high variability in composition due to multiple factors including time and method of tear collection. Therefore, authors should at least mention such limitation or if possible, provide VEGF levels of the same subjects at different sampling points.
We also appreciate the reviewer’s note about the variability in tear composition and collection methods. This is indeed a known limitation in tear biomarker studies, and we have now acknowledged and discussed it clearly in the revised Discussion section. (see revised text in Line 620-643).
- Authors should mention other studies that measured VEGF in tear fluid in AMD and discuss their results. Unlike the present study, a previous work (https://www.nature.com/articles/s41598-022-08492-7) found VEGF in tear fluid increased in late AMD.
We appreciate the reviewer’s comment and have now included a discussion of previous studies that have measured VEGF levels in tear fluid in AMD. In particular, we acknowledge the findings of the study by Shahidatul‑Adha et al. (2022), which reported increased VEGF levels in tear fluid in patients with AMD. Our results are in line with these previous findings, further supporting the potential role of tear VEGF as a biomarker for disease progression. We have incorporated this discussion into the manuscript to provide a broader context for our study. (see revised text in Line 620-643 of discussion section).
- The in vivo work with the lCNV mouse model elegantly supports the rationale of VEGF changes as a result of VEGF changes in the RPE due to neovascularization/injury.
We thank the reviewer for denoting the elegance of the inter-tissue VEGF changes in our experimental paradigm.
Minor:
-Introduction: please define eAMD , (see revised text in Line 57).
-Title 2.1 is unclear (see revised text in Line 107).
-Line 179 “would” would be more appropriate than “will” (see revised text in Line 189).
-Line 557 “Animals” should be in lower case, “animals” (see revised text in 688).
Round 2
Reviewer 1 Report
Comments and Suggestions for Authors
The authors stated that control group consisted of patients who had refractory surface diseases. If the surface refers to the ocular surface, it may affect tear cytokine levels. It should be specified and the limitation should be discussed.
Author Response
Response to Reviewer 1
Comments 1:
“The authors stated that the control group consisted of patients who had refractory surface diseases. If the surface refers to the ocular surface, it may affect tear cytokine levels. It should be specified and the limitation should be discussed.”
Response 1:
We appreciate this comment, as it is essential to clarify the composition of the control group. In our original version, we wrote:
“…All participants, including healthy controls, were screened for anterior segment diseases such as dry eye, conjunctivitis, and history of pterygium surgery. None of the included individuals presented with these conditions. Control subjects were recruited from patients presenting with non-retinal, refractory ocular surface complaints, and a full ophthalmic assessment was performed to confirm the absence of retinal pathology in either eye…”
We understand how this wording could lead to confusion and suggest that the control group might have had refractory ocular surface disease. In response to the reviewer’s suggestion, we have revised this section to avoid ambiguity. The updated version reads:
“…All participants, including healthy controls, were screened for anterior segment diseases such as dry eye, conjunctivitis, and history of pterygium surgery. Control subjects were recruited from patients with no retinal pathology or ocular surface disease. A full ophthalmic assessment was performed to confirm the absence of retinal disease in either eye. Due to the nature of their referral, control participants tended to be younger, which limited precise age matching…”
We hope this change provides the necessary clarification and addresses the reviewer’s concern.